# FROM EXPLICIT COT TO IMPLICIT COT: LEARNING TO INTERNALIZE COT STEP BY STEP

## ABSTRACT

When leveraging language models for reasoning tasks, generating explicit chain-of-thought (CoT) steps often proves essential for achieving high accuracy in final outputs. In this paper, we investigate if models can be taught to internalize these CoT steps. To this end, we propose a simple yet effective method for internalizing CoT steps: starting with a model trained for explicit CoT reasoning, we gradually remove the intermediate steps and finetune the model. This process allows the model to internalize the intermediate reasoning steps, thus simplifying the reasoning process while maintaining high performance. Our approach enables training a GPT-2 Small model to solve 20-by-20 multiplication with 99.5% accuracy while being 26 times faster than explicit CoT, whereas standard training cannot solve beyond 4-by-4 multiplication. Furthermore, our method proves effective on larger language models, such as Mistral 7B, achieving over 50% accuracy on GSM8K without producing any intermediate steps.

## 1 INTRODUCTION

A prevalent approach to improving the performance of language models (LMs) to perform complex reasoning tasks is chain-of-thought (CoT) reasoning, in which the LM generates explicit intermediate reasoning steps before arriving at a final answer (Nye et al., 2021; Wei et al., 2022). This method allows models to break down complex problems into simpler, manageable parts, thereby improving the accuracy of their final predictions. However, this explicit reasoning process can be computationally expensive, especially when the reasoning chain is long (Deng et al., 2023). Additionally, using explicit intermediate steps might not align with the intrinsic computational strengths of LMs (Lehnert et al., 2024): for instance, multi-digit multiplication is very easy for calculators but remains challenging for GPT-4 (Yang et al., 2023).

In this work, we examine the possibility of internalizing the reasoning process in the model's hidden states. We propose an approach, Stepwise Internalization, which begins with a model trained for explicit CoT reasoning. We then gradually remove the intermediate steps and finetune the model, forcing it to internalize the reasoning process. Once all intermediate steps are internalized, we achieve a model capable of full implicit CoT reasoning. Moreover, even in cases where the model does not have the capacity for full implicit CoT reasoning, this method still allows for shortening the reasoning chain while maintaining accuracy.

Our approach is an alternative to the approach proposed by Deng et al. (2023), which shares the goal of implicitly reasoning using the hidden states of transformers instead of relying on explicit CoT tokens. To teach the model to use hidden states for reasoning, that method employs a teacher model that performs explicit CoT reasoning, and then distills the teacher's hidden states into the student model's hidden states. In comparison, our approach is much simpler yet more effective.

Our approach demonstrates significant improvements over standard training methods. For instance, a GPT-2 Small model trained with Stepwise Internalization on multiplication can solve 20-by-20 multiplication problems nearly perfectly, while standard training without CoT struggles even with 4-by-4 multiplication. Furthermore, our method scales effectively to larger models, such as the Mistral 7B model (Jiang et al., 2023), achieving over 50% accuracy on the GSM8K dataset of grade-school math word problems (Cobbe et al., 2021), without producing any explicit intermediate steps, outperforming the much larger GPT-4 model without chain-of-thought reasoning, which only scores 44% when prompted to directly generate the answer.

It is important to note that our empirical evaluation focuses on specific reasoning tasks like multi-digit multiplication and grade-school math problems. While our results show the potential of Stepwise Internalization in these contexts, and the simplicity of the method makes it applicable to chain-of-thought approaches in a wide range of tasks, further research is needed to explore its efficacy across a broader range of tasks and more diverse CoT traces. Due to limitations in available computational resources, experiments on other tasks are out of scope for this work. This paper aims to lay the groundwork for this new approach and highlight its promise, while acknowledging that its full generalization is still under investigation.

The contributions of our work are as follows: First, we introduce Stepwise Internalization, a simple method for implicit CoT reasoning. Second, we demonstrate the effectiveness of internalizing intermediate hidden states via Stepwise Internalization. Third, we provide empirical results showing the superior performance of models trained with Stepwise Internalization on different reasoning tasks and model scales. Our code, data, and pretrained models are available at `https://anonymous.4open.science/r/Internalize_CoT_Step_by_Step`.

## 2 BACKGROUND: IMPLICIT CHAIN-OF-THOUGHT REASONING

Implicit chain-of-thought reasoning (implicit CoT, or ICoT) is a concept introduced by Deng et al. (2023), where during generation, the language model does not produce explicit intermediate reasoning steps in words. It is distinct from not using chain-of-thought reasoning (No CoT), in that explicit reasoning steps are allowed during training, enabling the ICoT model to learn the underlying reasoning approach from the supervision provided on the reasoning process. The key insight of Deng et al. (2023) is that intermediate reasoning steps serve two purposes in explicit CoT: they provide supervision during training to facilitate learning the task (Nye et al., 2021), and they act as a scratchpad during inference to assist in solving the task (Wei et al., 2022). However, the latter purpose can be fulfilled by utilizing the internal states of the model instead of explicit tokens.

As an illustrative example, consider using a language model to solve a multi-digit multiplication problem, such as $12 \times 34$. (The actual input reverses the digit order as `2 1 * 4 3` for consistency with Deng et al. (2023).) In the long multiplication algorithm, $12 \times 34$ is broken into:

$$12 \times 4 + 12 \times 30 = \underbrace{48}_{\text{reversed: 84}} + \underbrace{360}_{\text{reversed: 063}} \ .$$

In explicit CoT, the model is trained to predict these intermediate steps `8 4 + 0 6 3` before predicting the final answer `8 0 4` (408 reversed). Predicting these intermediate steps facilitates the model's ability to solve the task. (The intermediate steps are also reversed to make it easier for the model to predict (Shen et al., 2023).)

In both No CoT and implicit CoT settings, the model needs to directly predict the answer 408 from the input, bypassing the intermediate steps. This approach can make inference much faster for long reasoning chains, albeit at the cost of accuracy.

The primary difference between implicit CoT and No CoT lies in the use of intermediate reasoning steps as supervision during training. In Deng et al. (2023), a knowledge distillation approach was employed to distill explicit reasoning into implicit reasoning within the hidden states. This method involves training a teacher model to perform explicit CoT reasoning and then transferring this knowledge to a student model, which internalizes the reasoning process within its hidden states.

In the present work, we propose a far simpler yet more effective approach based on a kind of curriculum learning that we call Stepwise Internalization, which we detail in the next section.

## 3 STEPWISE INTERNALIZATION

Stepwise Internalization is a method designed to achieve implicit chain-of-thought reasoning by gradually removing intermediate reasoning steps during training. We define the input as $x$, the intermediate steps as $z = z_1, z_2, \cdots, z_m$, and the final output as $y$. A language model with parameters $\theta$ is first trained using the following loss function:

$$\min_{\theta} -\log P_{\theta}(y, z_{1:m} \mid x),$$

|  |  | Input |  |  |  |  |  | CoT |  |  |  |  |  | Output |  |  |  |
|---|---|---|---|---|---|---|---|---|---|---|---|---|---|---|---|---|---|
| Explicit CoT | Stage 0: | 2 | 1 | × | 4 | 3 | = | 8 | 4 | + | 0 | 6 | 3 | = | 8 | 0 | 4 |
|  | Stage 1: | 2 | 1 | × | 4 | 3 | = |  | 4 | + | 0 | 6 | 3 | = | 8 | 0 | 4 |
|  | Stage 2: | 2 | 1 | × | 4 | 3 | = |  |  | + | 0 | 6 | 3 | = | 8 | 0 | 4 |
|  | Stage 3: | 2 | 1 | × | 4 | 3 | = |  |  |  | 0 | 6 | 3 | = | 8 | 0 | 4 |
|  | Stage 4: | 2 | 1 | × | 4 | 3 | = |  |  |  |  | 6 | 3 | = | 8 | 0 | 4 |
|  | Stage 5: | 2 | 1 | × | 4 | 3 | = |  |  |  |  |  | 3 | = | 8 | 0 | 4 |
| Implicit CoT | Stage 6: | 2 | 1 | × | 4 | 3 | = |  |  |  |  |  |  | = | 8 | 0 | 4 |

Figure 1: Stepwise Internalization for Implicit CoT Reasoning. This figure uses $12 \times 34$ as an example. The training process consists of multiple stages. At Stage 0, the model is trained to predict both the full chain-of-thought (CoT) and the final output, which is the same as explicit CoT training. At Stage 1, the first CoT token is removed, and the model is finetuned to predict the remaining CoT tokens and the output. This process continues with each subsequent stage removing an additional CoT token. By Stage 6, all CoT tokens have been removed, and the model is trained to directly predict the output from the input, achieving implicit CoT reasoning. This gradual removal and finetuning process allows the model to gradually internalize the reasoning steps.

where $z_{1:m}$ denotes the sequence of intermediate steps $z_1, z_2, \cdots, z_m$.

At each step $t$ of the training process, we remove (up to) $s(t)$ tokens from the intermediate steps $z$. The updated loss function then becomes:

$$\min_{\theta} -\log P_{\theta}(y, z_{1+\min(s(t),m):m} \mid x).$$

There are multiple ways to parameterize $s(t)$. For instance, it might be based on a threshold of the loss value or a predefined schedule similar to learning rate schedulers used in optimizers. In this work, for simplicity, we use a linear schedule for removing tokens:

$$s(t) = \left\lfloor \Delta \frac{t}{T} \right\rfloor,$$

where $T$ is the total number of steps per epoch, and $\Delta$ is a hyperparameter controlling how many CoT tokens are removed per epoch. (Once $s(t)$ exceeds the number of actual chain-of-thought tokens, all tokens are removed.)

During initial experiments, we observed instability in the training process due to changes in the loss function over time. This instability arises for two primary reasons:

First, the optimizer commonly used in training language models, such as AdamW (Kingma & Ba, 2017; Loshchilov & Hutter, 2019), maintains estimates of second-order gradients. A sudden change in the loss function, caused by the removal of one more CoT token, results in abrupt changes in the second-order gradients. To address this issue, we reset the optimizer's state whenever an additional CoT token is removed.

Second, even if a model fits perfectly to the current loss when $s$ tokens are removed, transitioning to the next stage, where $s + 1$ tokens are removed, leads to a significant increase in the loss, as the model is not yet trained for this new setting. To mitigate this issue, we introduce a technique which we term "Removal Smoothing", where we add a small random offset to the original number of tokens to remove $s(t)$, such that:

$$s(t)^* = s(t) + o,$$

Table 1: Dataset statistics. The number of tokens is the median based on the GPT-2 tokenizer.

| Dataset | Size | | | # Input Tokens | | | # CoT Tokens | | | # Output tokens | | |
|---------|------|-----|------|-------|-----|------|-------|-----|------|-------|-----|------|
| | Train | Dev | Test | Train | Dev | Test | Train | Dev | Test | Train | Dev | Test |
| Parity | 808k | 1k | 1k | 159 | 159 | 159 | 128 | 128 | 128 | 1 | 1 | 1 |
| Coinflip | 32k | 1k | 1k | 930 | 930 | 930 | 533 | 533 | 533 | 1 | 1 | 1 |
| $4 \times 4$ Mult | 808k | 1k | 1k | 9 | 9 | 9 | 46 | 46 | 46 | 9 | 9 | 9 |
| $5 \times 5$ Mult | 808k | 1k | 1k | 11 | 11 | 11 | 74 | 74 | 74 | 11 | 11 | 11 |
| $7 \times 7$ Mult | 808k | 1k | 1k | 15 | 15 | 15 | 148 | 148 | 148 | 15 | 15 | 15 |
| $9 \times 9$ Mult | 808k | 1k | 1k | 19 | 19 | 19 | 246 | 246 | 246 | 19 | 19 | 19 |
| GSM8K | 378k | 0.5k | 1.3k | 40 | 51 | 53 | 19 | 21 | 24 | 2 | 2 | 2 |

where $o$ is a random variable with support of non-negative integers $\mathbb{Z}_{\geq 0}$, and its distribution is parameterized by another hyperparameter $\lambda$:

$$P(o) \propto \exp(-\lambda o).$$

When $\lambda = \infty$, $o = 0$ and we recover the version without Removal Smoothing. However, when $\lambda < \infty$, the model is trained to remove more than $s(t)$ tokens at step $t$ with a small probability, which helps smooth the transition into the next stage of removing $s(t) + 1$ tokens, reducing the abrupt jumps in the loss function.

Figure 1 illustrates the high-level idea of the Stepwise Internalization approach. The training process consists of multiple stages, where the model progressively learns to internalize reasoning steps by removing tokens from the CoT at each stage, eventually achieving implicit CoT reasoning.

## 4 EXPERIMENTAL SETUP

### 4.1 DATA

We evaluate our proposed Stepwise Internalization method on four reasoning tasks: the parity task from Graves (2017), the coinflip task from Wei et al. (2022), and multi-digit multiplication and grade-school math reasoning following the setup from Deng et al. (2023).

**Parity**  The parity task was introduced by Graves (2017), where the goal is to determine the parity of a sequence of numbers. In this task, the input is a sequence of 128 numbers.[1] A random number of positions (between 1 and 128) were randomly set to 1 or $-1$, and the rest were set to 0. The target is 1 if there is an odd number of 1's in the input and 0 otherwise. The chain of thought in this task is also a sequence of 128 numbers, with the $n$-th number being the parity of the subsequence from 1 to $n$. An example from this dataset is:

- **Input:** 1 -1 0 0 -1 0 -1 1 1 1 -1 [. . . 128 numbers in total]
- **CoT:** 1 1 1 1 1 1 1 0 1 [. . . ]
- **Target:** 0 [the number of 1's in input modulo 2]

**Coinflip**  We also evaluated the coinflip task introduced by Wei et al. (2022). This task asks the model to predict whether a coin is still heads-up after people either flip or do not flip the coin. The original task was constructed for evaluating few-shot prompting, but we adapted it by generating a training set of 32K examples with 64 names per example. The chain of thought in this case records whether the coin remains heads-up after each person's action. For example:

- **Input:** A coin is heads up. Yamy does not flip the coin. Nana flips the coin. [. . . 64 names in total] Is the coin still heads up?
- **CoT:** Yamy: yes Nana: no [. . . ]
- **Target:** no [whether the coin is still heads up in the end]

---

[1]Graves (2017) used 64 numbers, but we found it to be too simple to be solved even without CoT.

Table 2: Acc on parity & coinflip

| Task | Parity | Coinflip |
|------|--------|----------|
| Explicit CoT | 1.00 | 1.00 |
| Random | 0.50 | 0.50 |
| No CoT | 0.53 | 0.51 |
| ICoT-SI | 1.00 | 0.99 |

Table 3: Acc on GSM8K. [†]: 5-shot prompted

| Model | GPT-2 S | GPT-2 M | Mistral 7B | GPT-3.5[†] | GPT-4[†] |
|-------|---------|---------|------------|------------|----------|
| Explicit CoT | 0.41 | 0.44 | 0.68 | 0.62 | 0.91 |
| No CoT | 0.13 | 0.17 | 0.38 | 0.03 | 0.44 |
| ICoT-KD | 0.20 | 0.22 | - | - | - |
| ICoT-SI | 0.30 | 0.35 | 0.52 | - | - |

**Multi-digit multiplication.** We use two of the most challenging arithmetic tasks from BIG-bench (bench authors, 2023): 4-by-4 multiplication and 5-by-5 multiplication, as described by Deng et al. (2023). Given the effectiveness of Stepwise Internalization on these tasks, we extend our evaluation to 7-by-7 and 9-by-9 multiplication. The complexity of multiplication tasks grows significantly with the number of digits, as the program length grows quadratically with the number of digits (Dziri et al., 2024). We use the scripts and setup from Deng et al. (2023) to generate synthetic training data for our main experiments.[2]

**Grade school math.** We use the GSM8K dataset (Cobbe et al., 2021), with the augmented training data provided by Deng et al. (2023). Detailed dataset statistics are provided in Table 1.

## 4.2 BASELINES AND MODELS

We compare our method to the following baselines:

- **No CoT:** Models directly trained or finetuned without chain-of-thought supervision, except for GPT-3.5 and GPT-4 which are prompted with five-shot demonstrations.
- **Explicit CoT:** Models finetuned or prompted with explicit chain-of-thought reasoning. We use 5-shot prompting for GPT 3.5 and GPT-4 but full finetuning for other models.
- **ICoT-KD:** The implicit chain-of-thought via knowledge distillation method proposed by Deng et al. (2023).

Our proposed method, implicit chain-of-thought via Stepwise Internalization, is termed ICoT-SI. To verify the effectiveness of our approach across different model scales, we use pretrained models GPT-2 (Radford et al., 2019) and Mistral-7B (Jiang et al., 2023).

## 4.3 EVALUATION

Because the premise for implicit chain-of-thought methods is to approach the speed of no chain-of-thought and the accuracy of explicit chain-of-thought, we use two main evaluation metrics: First, we evaluate the accuracy of each method on the respective tasks of generating the final output. Second, we compare the inference speed of each method to the No CoT baseline. We measure speed, in examples per second, on an Nvidia H100 GPU with a batch size of 1. For ICoT-KD, we directly take numbers from Deng et al. (2023). However, due to hardware differences, we recompute speed relative to No CoT when speed numbers from ICoT-KD are not available.

## 5 RESULTS

Table 2, Table 3, Table 4, Table 5, and Figure 2 present the main results, where we compare Stepwise Internalization to baselines.

**Stepwise Internalization enables solving tasks not solvable by No CoT.** Stepwise Internalization (ICoT-SI) enables solving problems previously not solvable without using CoT. For example, as shown in Table 2, No CoT cannot solve either the parity or coinflip tasks, achieving accuracy around random guessing. In contrast, ICoT-SI can solve both tasks nearly perfectly.

---

[2]Following Deng et al. (2023), $K$-by-$K$ multiplication only considers $K$-digit numbers but not lower digits.

Table 4: Results on multiplication. ICoT-KD: Implicit CoT via knowledge distillation (Deng et al., 2023). ICoT-SI: Implicit CoT via Stepwise Internalization (this work). Acc measures the exact match accuracy. Speed measures the number of examples per second during inference using a batch size of 1, normalized by the speed of the corresponding No CoT model. [†]: 5-shot prompted.

| Model | $4 \times 4$ | | $5 \times 5$ | | $7 \times 7$ | | $9 \times 9$ | |
|---|---|---|---|---|---|---|---|---|
| | Acc | Speed | Acc | Speed | Acc | Speed | Acc | Speed |
| **GPT-2 Small (117M)** | | | | | | | | |
| Explicit CoT | 1.00 | 0.17 | 1.00 | 0.14 | 1.00 | 0.12 | 1.00 | 0.09 |
| No CoT | 0.29 | 1.00 | 0.01 | 1.00 | 0.00 | 1.00 | 0.00 | 1.00 |
| ICoT-KD | 0.97 | 0.67 | 0.10 | 0.71 | - | - | - | - |
| ICoT-SI | 1.00 | 1.02 | 1.00 | 1.00 | 0.95 | 1.00 | 0.99 | 1.00 |
| **GPT-3.5[†]** | | | | | | | | |
| Explicit CoT | 0.43 | 0.10 | 0.05 | 0.07 | 0.00 | 0.15 | 0.00 | 0.11 |
| No CoT | 0.02 | 1.00 | 0.00 | 1.00 | 0.00 | 1.00 | 0.00 | 1.00 |
| **GPT-4[†]** | | | | | | | | |
| Explicit CoT | 0.77 | 0.14 | 0.44 | 0.14 | 0.03 | 0.09 | 0.00 | 0.07 |
| No CoT | 0.04 | 1.00 | 0.00 | 1.00 | 0.00 | 1.00 | 0.00 | 1.00 |

Table 5: Comparison between ICoT-SI and MathGLM (Yang et al., 2023), where input and target digits are not reversed, consistent with the setup in the original MathGLM paper.

| Approach | Model Architecture | Layers | # Parameters | $4 \times 4$ Acc | $5 \times 5$ Acc |
|---|---|---|---|---|---|
| MathGLM | MathGLM-100M | 35 | 142M | 0.80 | 0.56 |
| | MathGLM-500M | 40 | 568M | 0.90 | 0.60 |
| | MathGLM-2B | 40 | 2.1B | 0.95 | 0.90 |
| ICoT-SI | GPT-2 Small | 12 | 117M | 0.97 | 0.88 |
| | MathGLM-100M | 35 | 142M | 0.99 | 0.93 |

On multiplication tasks (Table 4), No CoT fails even on $4 \times 4$ multiplication. In comparison, ICoT-SI enables a GPT-2 Small model to solve even $20 \times 20$ multiplication problems with an accuracy of over 99.5%, as shown in Figure 2a.

When compared to existing literature, ICoT-SI is also competitive. For instance, as shown in Table 4, while ICoT-KD fails to solve $5 \times 5$ multiplication using a GPT-2 Small model, ICoT-SI can solve $9 \times 9$ (and even $20 \times 20$) multiplication. Additionally, while ICoT-KD is slightly slower than No CoT due to the additional emulator model, ICoT-SI has the same speed as No CoT.[3] When compared to MathGLM (Yang et al., 2023) following the same setup of not reversing input and target digits, ICoT-SI outperforms a 2.1B parameter model using only 142M parameters, as shown in Table 5. Another related work, Shen et al. (2023), is able to train a GPT-2 Small model to solve up to $14 \times 14$ multiplication, whereas the approach proposed in this work can solve up to $20 \times 20$ (Figure 2a).

ICoT-SI enables internalizing CoT in a general way, making it applicable to tasks beyond arithmetic, such as grade-school math problems. On the GSM8K dataset (Table 3), ICoT-SI finetunes Mistral-7B to achieve an accuracy of 0.52, whereas GPT-4 only gets 0.44 without using intermediate steps.

**Stepwise Internalization lags behind explicit CoT in accuracy but is faster.** In terms of accuracy, implicit CoT methods still lag behind explicit CoT. For instance, a finetuned Mistral-7B model achieves an accuracy of 0.68 on GSM8K with explicit CoT, but ICoT-SI achieves only 0.51. However, implicit CoT methods offer significant speed advantages. As shown in Figure 2b, ICoT-SI is over 26 times faster than explicit CoT on $20 \times 20$ multiplication.

---

[3]The speed of ICoT-SI in Table 4 is not always exactly 1.00 due to randomness in hardware performance.

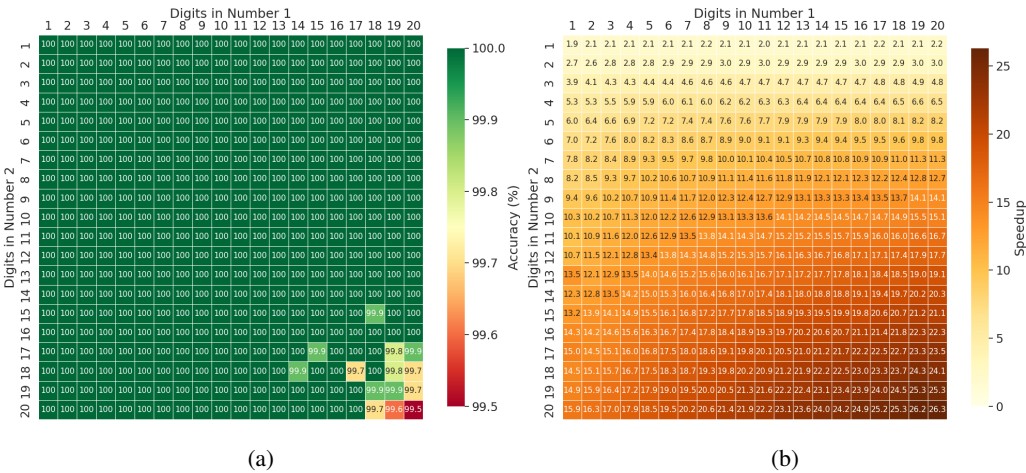

(a)                                                                                    (b)

Figure 2: (a) Accuracy of ICoT-SI on $m$-digit by $n$-digit multiplication tasks. (b) Speedup of ICoT-SI on $m$-digit by $n$-digit multiplication tasks compared to explicit CoT.

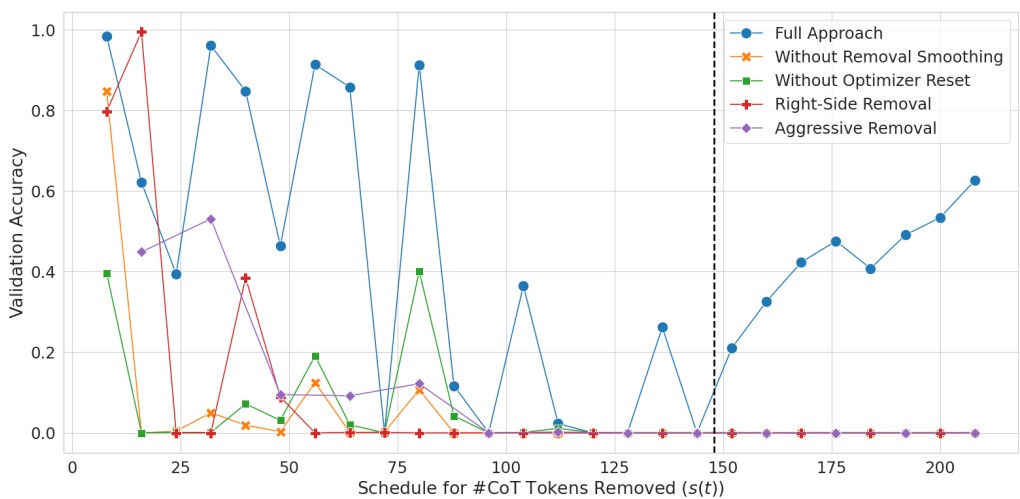

Figure 3: Accuracy during training for various ablations. This figure plots the validation accuracy versus the target number of removed CoT tokens during training ($7 \times 7$ Mult, GPT-2 Small). The black dashed vertical line indicates the point at which the schedule has removed all CoT tokens. The curves compare the following variants: "Full Approach"; "Without Removal Smoothing"; "Without Optimizer Reset"; "Right-Side Removal" (CoT tokens are removed from the end instead of the beginning); and "Aggressive Removal" (16 instead of 8 CoT tokens are removed per epoch).

Overall, our results demonstrate that Stepwise Internalization is an effective method for compressing chain-of-thought reasoning, offering a compelling trade-off between accuracy and speed. This makes it a valuable approach for reducing inference latency for tasks requiring long CoTs.

## 6 ANALYSIS

### 6.1 MECHANISTIC INTERPRETABILITY

To understand the internal algorithm used by the trained model, we conducted a probing analysis. We trained a probe to detect every digit in the explicit CoT for a $3 \times 3$ multiplication problem from

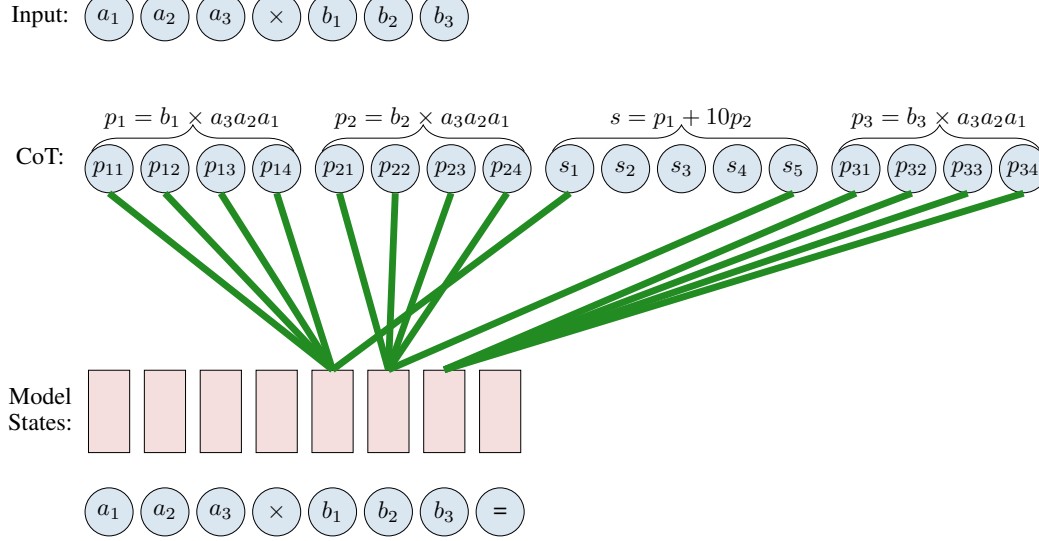

Figure 4: Mechanistic interpretability analysis of $3 \times 3$ multiplication. The top row displays the input digits; the middle row shows the CoT tokens in explicit CoT reasoning (partial products $p_1$, $p_2$, and $p_3$, and a partial sum $s = p_1 + 10p_2$); the bottom row represents the hidden states of a trained implicit CoT model. Connections indicate the earliest positions where each CoT token can be probed from the hidden states with over 95% accuracy.

the hidden states at each input position in an implicit CoT model.[4] Figure 4 shows the results. We observe that all partial products of the $3 \times 1$ multiplications have been computed at the earliest possible positions. For example, immediately after processing $b_1$, the hidden states already encode $p_1 = b_1 \times a_3 a_2 a_1$; similarly, after $b_2$, they encode $p_2 = b_2 \times a_3 a_2 a_1$. This suggests that the implicit CoT model internally replicates these partial product steps in explicit CoT.

However, the implicit CoT reasoning process doesn't fully replicate explicit CoT reasoning: the partial sum $s = p_1 + 10p_2$ cannot be reliably detected in any hidden state. This might be due to limitations of the probe, but it's also likely that the model doesn't compute the partial sum internally.[5] We suspect that the model first computes the partial products and then sums them in parallel, leveraging the language model's ability to sum sequences of numbers together (Chen et al., 2024). This might explain why a 12-layer GPT-2 Small can be trained to perform $20 \times 20$ multiplication.

## 6.2 ABLATION STUDIES

Figure 3 plots the validation accuracy versus the schedule for the number of CoT tokens removed during training for the $7 \times 7$ multiplication task. This figure compares the full approach to several ablated variants. Even for the full approach, there are fluctuations in the curve, and the validation accuracy briefly drops to zero at one point during training but eventually recovers. However, the ablated variants do not fully recover when accuracy drops.

**Removal smoothing.** As mentioned in Section 3, adding a small random offset $o$ to the number of removed tokens is crucial when the loss function changes due to the removal of more CoT tokens. The distribution of $o$ is parameterized by a hyperparameter $\lambda$, as introduced in Section 3. We use $\lambda = 4$ throughout this work, resulting in the distribution shown in Figure 5. In this distribution, 98% of the time, $o = 0$, but about 2% of the time, one or more additional tokens are removed. As shown in Figure 3, the "Without Removal Smoothing" curve fails to recover after the accuracy drops to zero at around $s(t) = 50$, whereas the full approach does much better.

---

[4]The probe is a one-layer MLP with a hidden dimension 16 times the input dimension. We trained one probe per layer, sharing it across all positions within the same layer by adding a trainable position embedding to the input hidden states. The accuracy per position is the maximum across all layers.

[5]We observed a similar phenomenon with $4 \times 4$ multiplication.

**Resetting the optimizer.** Another important technique for stabilizing training is resetting the optimizer when more tokens are removed. This avoids large estimates of second-order derivatives and stabilizes training. In Figure 3, the "Without Optimizer Reset" curve drops to zero around 100 steps and does not recover, showing the importance of resetting the optimizer during training.

**Removal side.** In our main experiments, CoT tokens are removed from the beginning (left side). Removing CoT tokens from the right side performs significantly worse, as shown by the "Right-Side Removal" curve in Figure 3. We suspect this is because internalizing tokens at the beginning is easier than internalizing tokens at the end. CoT tokens at the end depend on the earlier tokens, so internalizing them between the end of CoT and the beginning of the final answer, which only has a few positions, is more challenging. In contrast, internalizing tokens at the beginning allows distributing them across the entire input.

**Number of tokens removed per epoch.** The number of tokens removed per epoch ($\Delta$) significantly affects the training stability and speed. In the main experiments, we used $\Delta = 8$, which removes 8 tokens per epoch. A higher $\Delta$ value leads to faster training but risks not converging, as the model may not be able to keep up with the rapid changes in the loss function. For instance, when using $\Delta = 16$, the training fails to converge, as shown by the "Aggressive Removal" curve in Figure 3. Conversely, a lower $\Delta$ value is more likely to result in successful training but at a slower pace. Future work could explore adaptive $\Delta$ schedules based on loss values to balance speed and stability more effectively.

### 6.3 MODEL ARCHITECTURE

When comparing different model architectures for ICoT-SI in Table 5, we find that MathGLM with 35 layers outperforms GPT-2 Small with 12 layers, despite having a similar number of parameters. This result suggests that deeper models may be better for implicit reasoning, echoing the findings of Ye et al. (2024).

### 7 RELATED WORK

**No CoT approaches.** Several works in the literature focus on training language models to solve arithmetic tasks without outputting intermediate steps. MathGLM (Yang et al., 2023) demonstrated that with sufficient training data, including both lower-digit and higher-digit arithmetic task demonstrations, a 2 billion parameter LM can solve multi-digit arithmetic tasks without any intermediate steps. Compared to this work, Stepwise Internalization achieves higher accuracy in solving multi-digit multiplication with much smaller models, likely due to leveraging chain-of-thought supervision during training. Another notable work by Shen et al. (2023) showed that by mixing lower-digit and higher-digit multiplication demonstrations, even a GPT-2 Small can learn up to 14-digit multiplication. However, Stepwise Internalization does not require specially prepared training data with mixed task difficulties. Additionally, Stepwise Internalization is theoretically applicable to any reasoning task with CoT reasoning steps, as demonstrated by its effectiveness on grade-school math problems.

Also relevant is the work of Pfau et al. (2024), which shows that transformer language models can reason using filler tokens as an alternative to CoT tokens. They showed reasoning using these filler tokens improves a language model's expressivity. Our approach has the potential to be combined with their approach to solve even more challenging tasks.

**Internalizing CoT.** Our work is closely related to that of Deng et al. (2023) (ICoT-KD), which introduced the task of implicit CoT reasoning. ICoT-KD allows using CoT during training but not

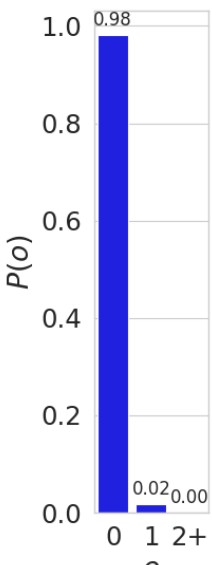

Figure 5: Distribution over Random Removal Offset $o$ in Removal Smoothing with $\lambda = 4$. The distribution is mostly concentrated at $o = 0$ with a probability of 0.98, and $o \geq 1$ has a probability of only 0.02. Despite this, the removal smoothing proves to be effective, as demonstrated in the ablation studies.

during generation, and it implements this via knowledge distillation to internalize the reasoning steps within hidden states. Compared to ICoT-KD, Stepwise Internalization has three advantages: First, it is simpler to implement as it does not require a teacher model. Second, while ICoT-KD internalizes reasoning into a single "column" of states (corresponding to the final input position), Stepwise Internalization allows the model to internalize reasoning across all input positions. Lastly, Stepwise Internalization achieves better accuracy compared to ICoT-KD.

Our work is also related to Context Distillation (Snell et al., 2022), which trains a model to produce the same output when conditioned on a scratchpad versus without it. Each stage of Stepwise Internalization can be viewed as a form of context distillation, where one CoT token is distilled into the model's internal states.

Another relevant work is Searchformer (Lehnert et al., 2024), which first trains a transformer to imitate A* search and then finetunes it on sampled shorter search traces. While Searchformer relies on sampling to find shorter traces, Stepwise Internalization forces the model to internalize steps.

## 8  LIMITATIONS

**Training costs.**  One limitation of the proposed approach is its high training cost due to the fine-tuning required when removing each set of CoT tokens. As discussed in Section 6.2, removing CoT tokens too fast leads to non-convergence. Therefore, the longer the CoT chain, the longer the training duration. For tasks like $N$-digit multiplication, where the reasoning chain length grows quadratically with $N$, training becomes expensive as $N$ increases.

**Instability.**  Another practical issue we observed is the instability of training with aggressive $\Delta$ values. For example, Figure 6 in Appendix B shows a case where the model could not recover from a drop in accuracy. Using lower $\Delta$ values generally leads to more stable training, but at the cost of longer training time. Identifying and addressing unstable dynamics early on, potentially by restarting training as suggested by Hu et al. (2024), could be a valuable improvement.

**Interpretability.**  Models trained using our approach lose interpretable intermediate steps. However, as shown in Section 6.1, it is still possible to interpret the internal hidden states of these models using probing techniques (Belinkov, 2018; Hewitt & Liang, 2019). Additionally, combining implicit and explicit CoT training could allow users to choose between interpretability and latency, providing flexibility based on the requirements of future tasks.

**Accuracy.**  Undoubtedly, explicit CoT still achieves higher accuracies compared to our approach to implicit CoT. However, our method enables a trade-off between latency and accuracy. Moreover, our results demonstrate the potential of leveraging hidden states for reasoning: even a GPT-2 Small model can be trained to solve $20 \times 20$ multiplication, despite having only 12 layers, far fewer than the number of reasoning steps in the CoT for $20 \times 20$ multiplication. When scaled to larger models with hundreds of billions of parameters and up to a hundred layers, such as GPT-3 (Brown et al., 2020), they could potentially solve even more challenging reasoning tasks without explicit CoT steps.

## 9  CONCLUSION AND FUTURE WORK

In this work, we introduced Stepwise Internalization, a novel approach for achieving implicit chain-of-thought reasoning in language models. By gradually removing intermediate CoT tokens and finetuning the model, we enable the internalization of reasoning steps incrementally. Our approach demonstrates significant improvements over existing methods, achieving high accuracy on up to $20 \times 20$ multiplication using GPT-2 Small and outperforming GPT-4 on GSM8K while not outputting any intermediate reasoning steps. Compared to explicit CoT methods, our approach can be up to 26 times faster while maintaining similar accuracies.

For future work, developing a mixed-mode approach that combines implicit and explicit CoT reasoning could potentially offer the best of both worlds, balancing accuracy, latency, and interpretability based on user preferences. Another promising direction is scaling Stepwise Internalization to larger models and more extensive training/pretraining setups, which could further enhance its effectiveness on a broader range of reasoning tasks.

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

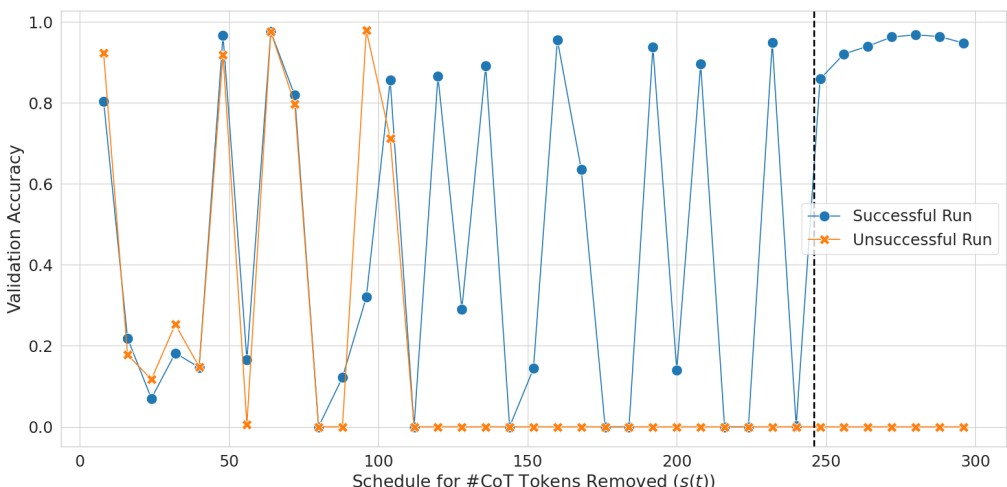

Figure 6: Validation Accuracy during Training for two different random seeds. This figure plots the validation accuracy as a function of the potential number of removed CoT tokens during training for the $9 \times 9$ multiplication task using GPT-2 Small and $\Delta = 8$. The two curves only differ in random seeds. The black dashed vertical line indicates the point beyond which all CoT tokens are removed.

## A  HYPERPARAMETERS

For all experiments, we use the AdamW optimizer (Loshchilov & Hutter, 2019), with $\lambda = 4$ and an effective batch size of 32 by default. For Mistral 7B, we use a batch size of 16 with a gradient accumulation of 2. For the multiplication tasks, we use a learning rate of $5 \times 10^{-5}$ and $\Delta = 8$. For GSM8K, we use a learning rate of $5 \times 10^{-5}$ and $\Delta = 1$ for GPT-2 Small and GPT-2 Medium, and a learning rate of $1 \times 10^{-5}$ and $\Delta = 8$ for Mistral 7B, with bfloat16 precision. Additionally, for GSM8K, we only consider sequences with 150 or fewer tokens for training and remove all CoT tokens when 39 or more tokens are scheduled to be removed. All experiments are run on a single H100 with 80GB of GPU memory for up to 200 epochs or 24 hours, whichever is reached first.

## B  STABILITY ISSUES FOR AGGRESSIVE REMOVAL

We found that using aggressive removal schedules (that is, bigger $\Delta$ values) can sometimes lead to unstable training dynamics. As one example, Figure 6 shows two different runs under identical configurations except for the random seed. One run was eventually able to solve the task after all CoT tokens were removed, whereas the other failed to solve the task after all CoT tokens were removed.

## C  ADDITIONAL EXPERIMENTS

**Keeping position IDs.**  As CoT tokens are removed, the position where the final output starts changes. We tried a variant where position IDs remain unchanged, meaning the position ID of the next token is used directly after removing a CoT token. Although this approach was more stable during training, its performance was similar to the current approach. For simplicity, we did not use this variant in our main experiments.

**Alternative CoT formats.**  Different valid reasoning paths can lead to the correct final answer for the same problem. We explored using a binary tree formatted CoT chain for the multiplication problems. This format decomposes an $N$-digit multiplication into a sequence of $N$-digit-by-1-digit multiplication problems, merges the results using sum operators, and continues merging until the final sum is computed. This program has a shorter description length, potentially making it easier

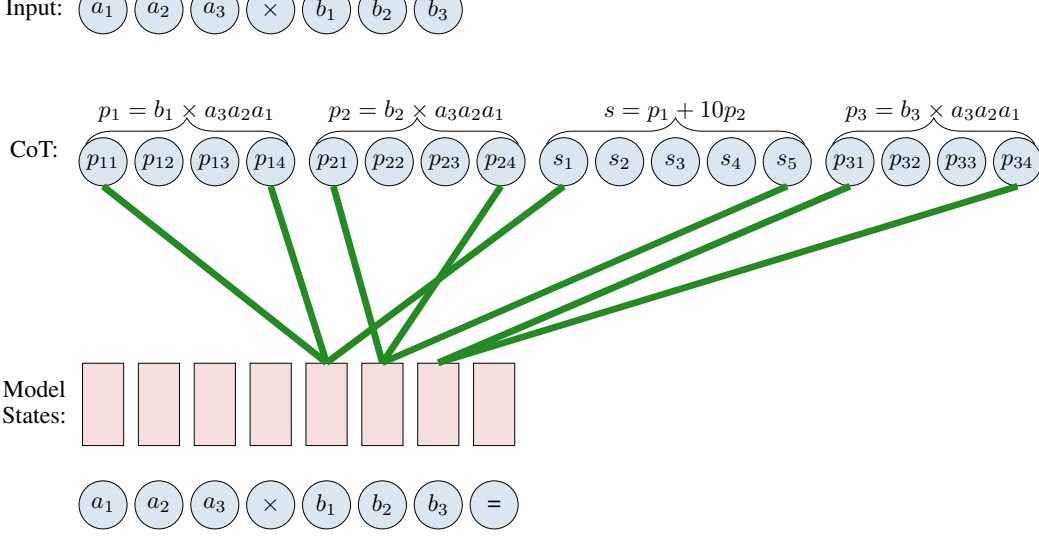

Figure 7: Mechanistic interpretability analysis of $3 \times 3$ multiplication on an explicit CoT model. The top row shows the input digits; the middle row depicts the CoT tokens used in explicit CoT reasoning (partial products $p_1$, $p_2$, $p_3$, and a partial sum $s = p_1 + 10p_2$); the bottom row represents the hidden states of a trained explicit CoT model. The connections indicate the earliest positions where each CoT token can be probed from the hidden states with an accuracy greater than 95%.

for transformers to learn (Dziri et al., 2024). However, its performance was similar to the current approach.

## D  DETAILS OF MECHANISTIC INTERPRETABILITY ANALYSIS

For the probing experiment, we used a one-layer MLP as the probe, with a hidden dimension set to 16 times the input dimension (input dimension = 768, corresponding to the hidden size). A separate probe was trained for each layer, using the hidden state at a given position as the input. To enable position-agnostic probing within a layer, we incorporated a trainable position embedding of the same size as the hidden dimension, allowing a single probe to be shared across all positions within the layer.

The probes were optimized using the AdamW optimizer with a learning rate of 5e-5 and a batch size of 32. Each probe was trained for 10 epochs on the training set. Probing accuracy was evaluated on the validation set, and results were aggregated across layers by selecting the maximum accuracy at each position. A threshold of 95% accuracy was used to generate the visualizations.

## E  ADDITIONAL MECHANISTIC INTERPRETABILITY ANALYSIS

As a control experiment, Figure 7 presents the mechanistic interpretability analysis performed on an explicit CoT model. Surprisingly, we observed that even in an explicit CoT model, which was not trained to internalize any CoT tokens, the internal hidden states can predict some CoT tokens. Specifically, the first and last digits of the partial products and partial sums are predictable at the earliest positions.

We hypothesize that this is because predicting these digits is relatively straightforward, and the probe likely learned the necessary patterns. For instance:

- Predicting the lowest digit of the first partial product ($p_{11}$) only requires computing $a_1 \times b_1$.

- Similarly, predicting the highest digit of the first partial product ($p_{14}$) largely depends on computing $a_3 \times b_1$, except in rare cases where a carry from lower digits alters the result.

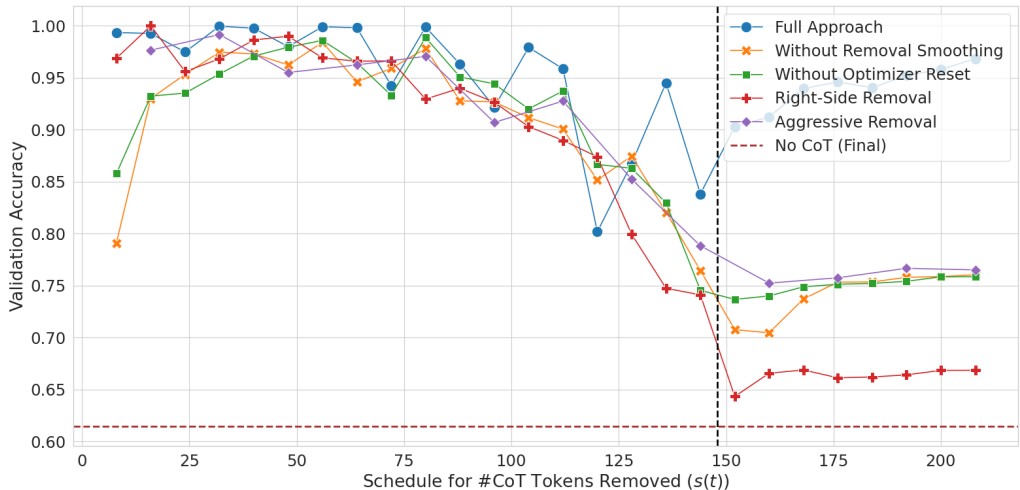

Figure 8: Token-level accuracy during training for various ablations. This figure shows the validation token-level accuracy versus the target number of removed CoT tokens during training ($7 \times 7$ Multiplication, GPT-2 Small). The brown dashed horizontal line represents the final token-level accuracy achieved by the No CoT approach, while the black dashed vertical line marks the point at which all CoT tokens have been removed according to the schedule. The curves compare the following variants of our approach: "Full Approach"; "Without Removal Smoothing"; "Without Optimizer Reset"; "Right-Side Removal" (removing CoT tokens from the end instead of the beginning); and "Aggressive Removal" (removing 16 instead of 8 CoT tokens per epoch).

For the partial sums, the situation is analogous:

- The lowest digit of the first partial sum ($s_1$) is equivalent to $p_{11}$, making it predictable from the same position.
- Similarly, the highest digit of the final partial sum ($s_5$) corresponds to $p_{23}$, which is also predictable from the position where $p_{23}$ is predictable.

When comparing the probing results of the explicit CoT model (Figure 7) with those of the implicit CoT model (Figure 4), the differences are clear. In the implicit CoT model, the hidden states contain information about the full partial product results, including $p_{12}$, $p_{13}$, $p_{22}$, $p_{23}$, $p_{32}$, and $p_{33}$—all of which are not predictable from the hidden states of the explicit CoT model.

This discrepancy highlights the key advantage of internalization: the training process compels the model to internalize intermediate computations, such as partial products, within its hidden states. This internalization enables the implicit CoT model to encode richer computational information, which is absent in the explicit CoT model despite having the same capacity.

## F    TOKEN ACCURACY

Figure 8 illustrates the token-level accuracy for various ablation variants of our proposed approach. Notably, for all variants, token-level accuracy remains above 90% during most of the training process (when fewer than 112 CoT tokens have been removed). However, as training progresses and all CoT tokens are removed, the token-level accuracy of the ablation variants drops to approximately 75% and does not recover. Qualitative analysis reveals that these variants often struggle with predicting the middle digits of the final result.

In contrast, the No CoT approach, despite training for over 24 hours, achieves a token-level accuracy of only 61% (represented by the brown dashed horizontal line). This difference underscores the effectiveness of our internalization approach in retaining high token-level accuracy throughout the training process, even under ablated conditions.

