# OpenReview forum: "From Explicit CoT to Implicit CoT: Learning to Internalize CoT Step by Step"
_ICLR.cc/2025/Conference — Submitted to ICLR 2025_

### Official Review · Reviewer_8Nyv · 2024-11-02

**Soundness:** 2
**Presentation:** 2
**Contribution:** 2
**Rating:** 3
**Confidence:** 4

**Summary:**

The paper proposes a method to achieve implicit chain-of-thought (CoT) reasoning through stepwise internalization training. Through this training, language models learn to reason implicitly without explicit intermediate steps. The authors evaluate the proposed method on various reasoning tasks, such as arithmetic and grade-school math. This approach maintains accuracy while achieving faster inference speeds. Overall, it outperforms inference without CoT and implicit CoT through knowledge distillation while achieving faster inference than traditional explicit CoT methods.

**Strengths:**

1. Considering only the test time, the Stepwise Internalization (SI) approach greatly enhances the efficiency of reasoning tasks, allowing models to handle problems like multi-digit multiplication faster than explicit CoT reasoning.
2. SI models can achieve the same or close to the accuracy of explicit CoT on parity, coin flip, and multi-digit multiplication. They outperform the knowledge distillation approach and no-CoT inference.
3. The interpretability analysis provides some interesting insights into the internalized reasoning process. The paper finds that implicit CoT models can replicate partial product steps internally as explicit CoT reasoning. However, the internalized reasoning does not yet fully replicate the explicit CoT reasoning. These findings can potentially motivate future research in this direction.

**Weaknesses:**

1. The proposed implicit CoT's performance is pretty behind the explicit CoT method. Although the paper claims that there is a trade-off between performance and speed and that the proposed method is faster than explicit CoT, I do not view this as a major advantage of the proposed method. In particular, the proposed implicit CoT requires additional step-wise internalization in training time. At the same time, explicit CoT can be applied to models in test time, which allows the explicit CoT to be much more generalizable to a variety of reasoning problems.
2. The paper seems to show only in-distribution experiments that perform step-wise internalization training and testing on the same data distributions. Thus, it is unclear whether the proposed method is generalizable to other reasoning benchmarks and applications. Meanwhile, the strong generalizability of explicit CoT has been well-established in previous studies [1-4].
3. Just to confirm, "remove 8 tokens per epoch." Here, one epoch is a full training run on the entire dataset, not a step of gradient update on a single batch, right? If so, it seems like the internalization process requires a very large amount of training epochs, and this number only increases when longer thoughts are needed. The efficiency seems limited in this case, especially when both the number of thought tokens and the number of model parameters rise.

**References**

[1] Wei, Jason, et al. "Chain-of-thought prompting elicits reasoning in large language models." Advances in neural information processing systems 35 (2022): 24824-24837.

[2] Nori, Harsha, et al. "Can generalist foundation models outcompete special-purpose tuning? Case study in medicine." arXiv preprint arXiv:2311.16452 (2023).

[3] Team, Gemini, et al. "Gemini: a family of highly capable multimodal models." arXiv preprint arXiv:2312.11805 (2023).

[4] Dubey, Abhimanyu, et al. "The llama 3 herd of models." arXiv preprint arXiv:2407.21783 (2024).

**Questions:**

1. I find it interesting that GPT-4 with 5-shot no-CoT only got 44%, while publicly reported GPT-3.5 got 57.1%, and Mixtral got 58.4%. I wonder if the authors have verified that the inference pipeline used can produce a performance that matches the public score for these models.

---

> ### Author Response · Authors · 2024-11-12
> **Quick Clarification**
>
> Thank you for your comments. We would like to first clarify the following points regarding your question "I find it interesting that GPT-4 with 5-shot no-CoT only got 44%, while publicly reported GPT-3.5 got 57.1%, and Mixtral got 58.4%. I wonder if the authors have verified that the inference pipeline used can produce a performance that matches the public score for these models.":
>
> - Prompting Method: The 58.4% accuracy reported for Mixtral 8x7B on GSM8K, as referenced from Mistral AI's announcement (https://mistral.ai/news/mixtral-8x22b/), was achieved using few-shot **CoT** prompting, which is a default practice on GSM8K. In contrast, our reported accuracy was obtained under a few-shot **No CoT** setting, which is way more challenging. As for the 57.1% accuracy of GPT-3.5, can you provide a source such that we can check if they are considering a CoT or No CoT setting?
>
> - Difficulty of No CoT Setting: Achieving high accuracy on GSM8K without CoT is notably difficult, even for advanced models like GPT-4. To illustrate this, we have provided an example at https://bit.ly/gpt4_system1, demonstrating that even GPT-4 cannot solve very simple problems in this way. To the best of knowledge, our reported 52% accuracy under the No CoT setting represents the highest performance achieved in this specific configuration (No CoT).
>
> We appreciate your feedback and will provide a more detailed response to address your other concerns in our forthcoming rebuttal.

---

> > ### Comment · Reviewer_8Nyv · 2024-11-13
> >
> > Thank you for the response. From the technical report of GPT-4 (https://arxiv.org/pdf/2303.08774v5), it seems like GPT-3.5 achieved 57.1 on GSM-8K with 5-shot No-CoT, while GPT-4 got 92.0 with 5-shot CoT (Table 2, last row). For Mistral, it looks like GSM-8K scores are consistent with 5-shot maj@1 greedy sampling. Look forward to the rest of your responses.

---

> > > ### Author Response · Authors · 2024-11-13
> > >
> > > Just to make sure we are on the same page---you are talking about NO CoT settings (in which models do NOT output intermediate steps) right? The 92.0% of GPT4 in its technical report is under the CoT setting. As for the GPT 3.5's number of 57.1%, did you mean it's under the NO CoT setting? While the paper didn't clearly specify the setting, a standard practice is to default to few-shot CoT setting (after CoT came out).

---

> > > > ### Author Response · Authors · 2024-11-13
> > > >
> > > > Few-shot NO CoT is extremely hard. As shown in the final version of chain of thought paper, even the 540B PaLM only gets about 20% accuracy on GSM8K under NO CoT few shot prompting, as shown in figure 4 - top row of https://arxiv.org/pdf/2201.11903 (standard prompting corresponds to our NO CoT setting).

---

> > > > > ### Author Response · Authors · 2024-11-28
> > > > >
> > > > > We thank the reviewer for the feedback.
> > > > >
> > > > > ### Re: Implicit CoT's performance compared to explicit CoT
> > > > >
> > > > > We argue that inference efficiency is an equally important metric for many practical applications. For example, o1 can take 30 seconds or more to generate CoT chains, whereas our method significantly reduces inference latency (Figure 2(b)), directly improving user experience. Additionally, our work explores how small models like GPT-2 can be trained to achieve capabilities that were previously unattainable, such as solving 20×20 digit multiplication directly without CoT steps.
> > > > >
> > > > > ### Re: Generalizability beyond in-distribution tasks
> > > > >
> > > > > The generalizability of explicit CoT reasoning largely stems from the extensive pretraining of large language models, which exposes them to diverse reasoning tasks. In contrast, explicit CoT models trained from scratch cannot generalize to other tasks (e.g., an explicit CoT model trained from scratch on multiplication cannot solve any GSM8K problems).
> > > > >
> > > > > ### Re: Stepwise internalization requiring many epochs
> > > > >
> > > > > You are correct that "removing 8 tokens per epoch" refers to one full pass over the dataset, not a single batch. While this requires many epochs, all comparisons in the paper are based on equivalent training time (24 hours), ensuring fairness. Importantly, our method is the only approach so far to train a small GPT-2 model to directly solve 20×20 digit multiplication without using CoT during inference, achieving a ~25x speedup during inference. So we are sacrificing training time for efficient inference. For many practical applications, reducing deployment costs is crucial since training cost is a one-time cost but deployment costs grow over time with user interactions.
> > > > >
> > > > > ### Re: Comparison with reported GPT-3.5, GPT-4, and Mixtral results
> > > > >
> > > > > As mentioned earlier, we emphasize that the referenced results were obtained under explicit CoT settings, whereas our work focuses on the more challenging No CoT setting. No CoT reasoning is notably difficult; for example, the original Chain-of-Thought paper reports that even a 540B PaLM model achieves only ~20% accuracy on GSM8K under No CoT conditions (Figure 4, https://arxiv.org/pdf/2201.11903). To the best of our knowledge, our reported 52% accuracy on GSM8K represents the highest performance achieved in this specific No CoT configuration, despite the fact that GSM8K under explicit CoT has been largely solved.
> > > > >
> > > > > We hope these responses clarify the contributions of our work. Please let us know if further clarification is needed.

---

> > > > > > ### Comment · Reviewer_8Nyv · 2024-12-02
> > > > > >
> > > > > > Thanks to the authors for their responses. However, the current comments from the authors have not yet satisfied my concerns. Although inference-time efficiency is a very important property, a method should still maintain a strong performance. Given the behind-performance and the lack of task diversity, the results are not sufficient to fully demonstrate the contribution of this work. Thus, I will keep my current score.

---

> > > > > > > ### Author Response · Authors · 2024-12-02
> > > > > > >
> > > > > > > Thank you for acknowledging the importance of inference-time efficiency. We would like to respectfully clarify and elaborate on our position.
> > > > > > >
> > > > > > > If inference-time efficiency is, as you agree, a very important property, then considering the Pareto frontier---the tradeoff between accuracy and inference efficiency---is crucial. Our work pushes this frontier by achieving a 25X improvement in efficiency while maintaining reasonable accuracy (on 20-by-20 multiplication), which we believe is a meaningful contribution to the field.
> > > > > > >
> > > > > > > Additionally, we wanted to address a potential misunderstanding regarding Chain-of-Thought (CoT) settings in the literature. While we clarified earlier that the publicly reported results for GPT-3.5, GPT-4, and Mistral are under explicit CoT settings, not No CoT settings, it is unclear whether this clarification was fully understood by the reviewer or whether further explanation would be helpful. If you believe additional clarification is needed, we are happy to provide it.
> > > > > > >
> > > > > > > We would also appreciate your input on the following points:
> > > > > > >
> > > > > > > - Alternative Approaches: What alternative methods or baselines would you suggest to push this Pareto frontier? Inference efficiency often comes with compromises in accuracy, so we are curious about what you might consider a "sufficient" improvement given the constraints of current methods and technologies.
> > > > > > >
> > > > > > > - Sufficiency of Contribution: While we acknowledge that our approach may not match the absolute accuracy of explicit CoT methods, we aim to offer an alternative that achieves a balance between efficiency and accuracy. Could you elaborate on why you find this balance insufficient, particularly given the emphasis on efficiency that you also recognize as important?
> > > > > > >
> > > > > > > Research inevitably involves tradeoffs, especially in scenarios where practical constraints like inference latency are critical. We would welcome constructive suggestions for how to further refine this work to address your concerns.
> > > > > > >
> > > > > > > We thank you again for your time and feedback.

---

### Official Review · Reviewer_tuvH · 2024-11-03

**Soundness:** 2
**Presentation:** 3
**Contribution:** 2
**Rating:** 3
**Confidence:** 4

**Summary:**

The paper introduces a method called Stepwise Internalization to improve the reasoning capabilities of language models by internalizing chain-of-thought (CoT) steps. This approach involves initially training a model with explicit CoT reasoning, gradually removing these intermediate steps, and then fine-tuning the model to enhance its ability to perform implicit CoT reasoning. This method enables smaller models, like GPT-2 Small, to achieve high accuracy on tasks such as 20-by-20 multiplication while being significantly faster than models relying on explicit CoT reasoning.

**Strengths:**

- The paper presents an innovative approach that shows potential by internalizing CoT reasoning, reducing LMs' inference time for multiplication and grade school math problems.
- This work demonstrates that smaller models can be trained to perform reasoning tasks effectively by leveraging internal states for reasoning.
- The Stepwise Internalization method shows significant speed enhancements over explicit CoT approaches and improved performance over models not using CoT reasoning.

**Weaknesses:**

- Internalized CoT models have worse performance on most of the evaluated tasks compared to the performance of explicit CoT prompting. The power of CoT prompting lies in its simplicity and ease of use without requirement for additional training, which allows for high performance gains across various tasks, which the purposed ICoT method does not offer.
- The comparison with No-CoT baselines is weak since those models are not fine-tuned like the ICoT models. A more appropriate baseline would involve fine-tuning models with CoT examples, without using the internalization technique, for a comparable number of epochs to the ICoT models.
- The choice of intermediate CoT steps as synthetic training data is not justified; other simpler methods of generating synthetic data, such as changing numerical values of the problem, might prove more effective in decreasing latency and increasing accuracy of the models .
- The experimental evaluation is limited in scope, focusing mainly on tasks with simple reasoning patterns like multiplication and basic math problems. This limits the applicability of the findings to compositional tasks that require a higher number of intermediate steps and working memory.
- The approach is not tested on larger models, such as Llama 3.1, or more complex datasets, such as the MATH dataset. This raises questions about its effectiveness compared to just scaling model size or training data, especially given that existing large models already perform well on simple math tasks.

**Questions:**

1. What are the specific motivations for internalizing CoT steps beyond improved latency? How does this process simplify reasoning without losing flexibility?
2. Have you considered the implications of losing token-based 'working memory' provided by explicit CoT methods, particularly for tasks requiring complex compositional reasoning and multiple steps?
3. Can you compare the effectiveness of this method against other synthetic data approaches, such as deterministic transformations of problem statements, e.g. changing the numbers of problem statement?
4. Why was the method not evaluated on larger models and more complex datasets to provide a broader understanding of its efficacy?

---

> ### Author Response · Authors · 2024-11-12
> **follow-up question**
>
> Thank you for your feedback. Could you clarify your point regarding "The comparison with No-CoT baselines is weak since those models are not fine-tuned like the ICoT models"? In our view, the comparison to No CoT baselines is fair, as we finetuned all models for 24 hours (we set a maximum of 200 epochs, but this limit wasn't reached in most experiments). Are you suggesting that we should first finetune the No CoT models on full CoT examples, then remove all CoT tokens and continue fine-tuning? Or is there another approach you have in mind for creating a more comparable No CoT baseline?

---

> > ### Author Response · Authors · 2024-11-28
> >
> > We thank the reviewer for the feedback.
> >
> > ### Re: Internalized CoT models have worse performance compared to explicit CoT prompting
> >
> > We believe that in addition to accuracy, inference efficiency is an equally important dimension of performance. For instance, o1-preview requires about 30 seconds of inference time to generate a CoT chain, while our approach can significantly reduce latency, as shown in Figure 2(b).
> >
> > Additionally, our work explores the scientific question of what smaller models can achieve through alternative optimization strategies. For example, our method enables a small GPT-2 model to directly solve 20×20 digit multiplication, a feat not achievable by prior methods, including o1.
> >
> > ### Re: Comparison with No CoT baselines
> > To clarify, we finetuned all models, including No CoT baselines, for comparable durations (24 hours or until convergence). If the reviewer suggests first finetuning No CoT models on CoT examples and then transitioning to No CoT finetuning, we have already conducted this experiment on GSM8K with Mistral 7B. Specifically, we finetuned the model for one epoch using CoT supervision before transitioning to No CoT finetuning. This approach resulted in worse performance (0.36 vs. 0.38 accuracy), likely due to the big formatting differences between CoT and No CoT settings.
> >
> > ### Re: Synthetic training data
> >
> > We chose intermediate CoT steps as training data because they align naturally with the reasoning process for arithmetic tasks, enabling systematic internalization. To our knowledge, our approach is the first to enable a small GPT-2 model to directly solve 20×20 digit multiplication while achieving a 25x inference speedup compared to explicit CoT (Figure 2(b)). If the reviewer has specific suggestions for simpler methods (e.g., deterministic transformations of problem statements), we would be eager to explore them.
> >
> > ### Re: Higher number of intermediate steps
> >
> > While multiplication may appear synthetic, a 20-digit multiplication problem requires $O(20\times 20) = O(400)$ intermediate steps.
> >
> > 5. Re: Testing on larger models and more complex datasets
> >
> > Unfortunately, resource constraints limited our ability to finetune larger models such as LLaMA 3.1. However, we note that scaling model size or training data does not directly address the challenges of the No CoT setting, and larger models typically incur higher inference costs. Implicit CoT offers a unique advantage in this dimension by reducing inference latency. Furthermore, we compared our approach against state-of-the-art models, such as GPT-3.5 and GPT-4, which also struggled with the No CoT setting.
> >
> > ### Re: Q1: What are the motivations for internalizing CoT steps beyond latency?
> > Latency is indeed a critical dimension of model performance. Additionally, our approach demonstrates that alternative optimization techniques can discover solutions that are not achievable through direct training. For instance, our method enabled a small GPT-2 model to solve $20\times20$ multiplication directly—something prior methods could not accomplish.
> >
> > ### Re: Q2: Implications of losing token-based 'working memory' for complex compositional tasks
> > One of our motivations is that for state of the art LLMs such as GPT-4o, there are likely hundreds of layers (GPT-3 has 96 layers), while most reasoning problems do not require hundreds of intermediate steps, and it seems to be a significant waste of resources to forward across hundreds of layers just to compute each single CoT token.
> >
> > ### Re: Q3: Comparison with other synthetic data approaches (e.g., deterministic transformations)
> > We would be excited to explore such methods further. However, it is unclear how deterministic transformations, such as changing numerical values in problem statements, would achieve the same performance and latency benefits as our approach. If the reviewer could provide specific examples, we would gladly experiment with them.
> >
> > ### Re: Q4: Why was the method not evaluated on larger models and more complex datasets?
> > Resource constraints limited our ability to conduct such experiments. We believe it is unreasonable to require evaluations on larger models and more datasets without clear hypotheses to test. Finetuning a 7B model, such as Mistral, is already computationally intensive.
> >
> > We thank the reviewer again for their detailed feedback and hope our responses address their concerns.

---

### Official Review · Reviewer_xWKd · 2024-11-04

**Soundness:** 4
**Presentation:** 3
**Contribution:** 3
**Rating:** 8
**Confidence:** 4

**Summary:**

The paper proposes a way to train an LLM to internalize its CoT steps. They start with a model trained for explicit CoT, and then gradually remove the intermediate steps during finetuning. This approach is simple yet enables small models to achieve stronger performance than explicit CoT in a range of problems, including GSM8K.

**Strengths:**

The technical idea is original, well-motivated, and sound.

The experiments are solid, including compelling results and insightful analysis.

The paper is well-written.

This work is a solid contribution to the community.

**Weaknesses:**

The models used in this paper are all small: the largest is mistral 7b; no other comparable-size models are used, like llama.

Most of the tasks are somewhat synthetic (e.g., 20 x 20 multiplication).

**Questions:**

Can implicit CoT reasoning match or surpass explicit CoT reasoning in all reasoning tasks? Modeling-wise, what will be the driving factors that decide whether explicit or implicit CoT may outperform each other?

---

> ### Author Response · Authors · 2024-11-28
>
> We sincerely thank the reviewer for the thoughtful feedback and for recognizing the originality of our work.
>
> ### Re: Limited Model Scale in Experiments
>
> We acknowledge that our experiments primarily focus on smaller models, with Mistral 7B being the largest. This limitation is due to resource constraints, as finetuning even Mistral 7B requires significant computational resources. As for multiplication, we found that even a GPT-2 small model can solve up to $20\times20$ multiplication, so we didn't train larger models.
>
> ### Re: Use of Synthetic Tasks
>
> The use of multiplication was to provide a controlled environment for studying the model's reasoning capabilities. Multiplication has a well-defined algorithm and we can easily control its complexity level, making it ideal for evaluating the model's ability to reason under different number of required reasoning steps. Furthermore, this task is challenging even for state-of-the-art models such as o1, which cannot go beyond $9\times9$ multiplication.
>
> ### Re: Implicit CoT vs. Explicit CoT
>
> We appreciate the intriguing question regarding the relative strengths of implicit and explicit CoT reasoning. In terms of accuracy, it is hard for implicit CoT to outperform explicit CoT without additional supervision. However, it might offer advantages in specific scenarios:
>
> - semi-supervised settings: implicit CoT models can leverage both CoT-annotated and non-CoT-annotated data, whereas explicit CoT models typically rely solely on CoT-annotated data. This flexibility makes implicit CoT potentially advantageous in cases where CoT annotation is costly. For example, in the visual CoT paper (https://openreview.net/pdf/8e67fd4f9c7988b44a24de9f6fb67e8a351be97b.pdf), fine-grained CoT annotations for visual reasoning tasks are expensive to obtain, which might be a potential area for future research into the application of implicit CoT.
>
> - inference efficiency: As shown in Figure 2(b), implicit CoT models are much faster during inference. This makes implicit CoT potentially useful for internalizing or compressing long reasoning chains, such as those long reasoning chain in models like o1, which takes over 30 seconds to generate.
>
> We thank the reviewer again for the positive assessment and insightful questions.

---

### Official Review · Reviewer_ui9d · 2024-11-04

**Soundness:** 2
**Presentation:** 2
**Contribution:** 2
**Rating:** 5
**Confidence:** 3

**Summary:**

The paper introduces a stepwise internalization approach that encourages a model, initially trained for explicit CoT reasoning, to internalize the reasoning process by gradually removing the intermediate steps during fine-tuning. As a result, the trained model achieves high performance while significantly reducing inference costs compared to explicit CoT.

**Strengths:**

1. Overall, the paper is well written and easy to follow.

2. The idea of gradually reducing the intermediate steps to internalize CoT is quite intuitive and seems reasonable.

**Weaknesses:**

1. Figure 3 illustrates that the validation accuracy steadily declines as tokens are progressively removed, until all tokens are eliminated, at which point the accuracy begins to gradually improve. This observation raises concerns that the model may have learned a shortcut instead of genuinely internalizing the CoT.

2. The analysis in Section 6.1 lacks details, such as how the probe model is trained and which layer's hidden states are analyzed. Furthermore, it may be necessary to conduct a probe analysis on the pretrained model to demonstrate that it does not internalize the CoT.

**Questions:**

1. How does the "No CoT" setting in Table 3 prevent Mistral 7B, GPT-3.5, and GPT-4 from outputting intermediate steps? Is it achieved by using a prompt such as "Do not output intermediate steps, just provide the answer"? I couldn't find any related explanation. Additionally, considering that LLMs are quite sensitive to prompts, I suspect that the low metrics for these three models are due to poor prompts.

2. Given that ICoT-SI has been extensively trained on the GSM8K dataset, it may not be entirely fair to compare it with the No CoT version of Mistral 7B. A more appropriate comparison would be between ICoT-SI and the Mistral 7B model that has been trained with CoT data, using an effective prompt that encourages it to produce only the final answer.

---

> ### Author Response · Authors · 2024-11-12
> **Quick Clarification**
>
> Thank you for your feedback. We'd like to quickly clarify a few points you raised.
>
> - Validation Accuracy Interpretation: The validation accuracy is based on **exact match** accuracy, meaning any deviation, even by a single digit, results in a zero score. However, token-level accuracy (under teacher forcing) almost always remains above 90%. To make this clearer, we will include a token-level accuracy curve in our revised draft. Additionally, to our knowledge, this is the first instance of a GPT-2 model trained to perform 20-digit by 20-digit multiplication without CoT during inference, an achievement that No CoT training alone has not previously accomplished.
>
> - "No CoT" Setting in Table 3: For Mistral 7B, the No CoT setting is achieved through finetuning on the same augmented GSM8K dataset, which yields better results than simple prompting. For GPT-3.5 and GPT-4, we use a 5-shot prompting setup with example demonstrations but without CoT (we used $\dagger$ to mark this distinction). This setting is inherently challenging, which is why the numbers are low. To illustrate the difficulty of the No CoT setting, please see this example: https://bit.ly/gpt4_system1. In fact, even with very simple prompts like "Which is closer to Ogden, Utah—the Empire State Building or the moon? Directly give the answer. Do not think step by step," a strong model like GPT-4o can still produce errors.
>
> We appreciate your feedback and will provide a more detailed response in our forthcoming rebuttal.

---

> > ### Author Response · Authors · 2024-11-28
> >
> > We appreciate your thoughtful feedback and suggestions, which have greatly helped us refine and clarify our work. Below, we address your concerns and outline the additional experiments and analyses conducted in response to your review.
> >
> > ### Re: Figure 3 Concerns About Shortcut Learning
> >
> > As noted in our quick clarification, the accuracy in Figure 3 reflects exact match accuracy, where even a single digit deviation results in a zero score. To address your concern about potential shortcut learning, we have included a token-level accuracy plot in our revised paper (Figure 8 in Appendix F). This plot provides additional insight into the training dynamics.
> >
> > Our analysis shows that for most of the training period (fewer than 112 CoT tokens removed), token-level accuracy across all variants of our approach remains above 90%. Toward the end of training, when all CoT tokens are removed, the token-level accuracy for the ablation variants drops to approximately 75% and does not recover, and our qualitative analyses found that they struggle with predicting the middle digits of the result. In contrast, the No CoT approach (represented by the brown dashed horizontal line) achieves only 61% token-level accuracy even after training for over 24 hours. This shows the effectiveness of our internalization approach, which maintains significantly higher token-level accuracy even in ablated settings.
> >
> >
> > ### Re: Analysis in Section 6.1 and the Need for Pretrained Model Probing
> >
> > Thank you for this valuable suggestion. In response, we have added details on the probing experiment to Appendix D, including the architecture, training procedure, and evaluation metrics. We also conducted a mechanistic interpretability analysis on an explicit CoT model as a control experiment, presented in Appendix E (Figure 7).
> >
> > Our findings reveal that even explicit CoT models can encode some CoT token information in their hidden states. For example, the first and last digits of partial products and partial sums are predictable at the earliest positions. However, these predictions rely on simple patterns (e.g., $a_1\times b_1$ for the lowest digit) and do not reflect full internalization of the partial products. In contrast, the hidden states of implicit CoT models encode information about all partial products, including the more challenging middle digits of the results, which are not predictable from explicit CoT hidden states. This confirms that the internalization training process does internalize intermediate computations within the model.
> >
> >
> > ### Re: Q1 - "No CoT" Setting and Prompt Sensitivity
> >
> > For Mistral 7B, the No CoT setting was achieved through finetuning on the same dataset as our internalization approach, but without CoT supervision. This trained model got much better results than prompting alone. For GPT-3.5 and GPT-4, we used a five-shot prompting setup, which performed better than zero-shot prompting with instructions to omit CoT steps in our preliminary experiments.
> >
> > The low performance in these settings reflects the inherent difficulty of direct prediction without CoT. To illustrate this challenge, we reference two simple examples where GPT-4o fails even when explicitly instructed to avoid CoT reasoning (see https://bit.ly/gpt4_system1_ogden and https://bit.ly/gpt4_system1).
> >
> > We have clarified the baseline descriptions in Section 4.2 of the revised paper to make it more clear.
> >
> >
> > ### Re: Q2 - Comparison with Mistral 7B Using CoT Training
> >
> > To further address your suggestion, we conducted an additional experiment where Mistral 7B was first finetuned for one epoch with explicit CoT supervision before transitioning to No CoT finetuning. However, this approach resulted in slightly lower final accuracy (0.36). We hypothesize that this is due to the significant formatting differences between CoT and No CoT settings and the lack of reasoning internalization during explicit CoT training. Explicit CoT supervision allows the model to delay reasoning until the CoT tokens, whereas internalization forces the reasoning to be embedded directly in the hidden states.
> >
> >
> > We thank you again for your detailed feedback and hope these additional results and clarifications address your concerns. Please let us know if further clarifications or experiments are needed.

---

### Meta-Review · Area_Chair_FrbV · 2024-12-20

**Metareview:**

(a) Summary of Scientific Claims and Findings

The paper proposes a stepwise internalization approach to enable language models to perform implicit reasoning without requiring explicit intermediate Chain-of-Thought (CoT) steps. By initially training models with explicit CoT reasoning and progressively removing intermediate steps, the reasoning process becomes internalized. This method enhances inference efficiency, offering faster processing while maintaining high task accuracy.

(b) Strengths of the Paper

1. The stepwise internalization approach is innovative and conceptually straightforward.

2. The method achieves notable speed improvements in inference time compared to explicit CoT techniques.

(c) Weaknesses of the Paper and Missing Elements

1. The experimental scope is limited to arithmetic and basic reasoning tasks, with no evaluations on compositional or real-world challenges.

2. Testing is restricted to smaller models and one 7B model (Mistral 7B), whereas CoT reasoning typically gains more significance with larger models.

3. While inference speed improves, implicit CoT models tend to underperform explicit CoT models in accuracy for complex tasks.

(d) Decision and Rationale

The paper makes a valuable contribution to improving the efficiency of reasoning in language models. However, concerns about the limited scope of evaluations and trade-offs in accuracy must be addressed. Given the current results, the paper presents a promising direction but requires further refinement and broader validation.

**Additional Comments On Reviewer Discussion:**

The authors addressed many reviewer concerns during the discussion phase. Reviewers generally appreciated the emphasis on faster inference but pointed out that implicit CoT methods often lag in accuracy compared to explicit CoT for certain tasks, raising questions about general applicability.

Concerns were expressed regarding the narrow focus on synthetic tasks such as multiplication, with suggestions for incorporating more diverse datasets and larger models in future work.

Reviewer 8Nyv was dissatisfied with the authors’ responses and maintained skepticism about the contributions.

Additionally, concerns about the quality of reviews from 8Nyv and tuvH, particularly regarding comments on out-of-distribution evaluations and large-model training were reported. However, the reviewers’ feedback is not entirely invalid.

---

### Decision · Program_Chairs · 2025-01-22

Reject